# Overactivated transport in the localized phase of the superconductor-insulator transition

V. Humbert [1,2], M. Ortuño [3✉], A. M. Somoza [3], L. Bergé[4], L. Dumoulin[4] & C. A. Marrache-Kikuchi [4✉]

Beyond a critical disorder, two-dimensional (2D) superconductors become insulating. In this Superconductor-Insulator Transition (SIT), the nature of the insulator is still controversial. Here, we present an extensive experimental study on insulating $Nb_xSi_{1-x}$ close to the SIT, as well as corresponding numerical simulations of the electrical conductivity. At low temperatures, we show that electronic transport is activated and dominated by charging energies. The sample thickness variation results in a large spread of activation temperatures, fine-tuned via disorder. We show numerically and experimentally that this originates from the localization length varying exponentially with thickness. At the lowest temperatures, there is an increase in activation energy related to the temperature at which this overactivated regime is observed. This relation, observed in many 2D systems shows that conduction is dominated by single charges that have to overcome the gap when entering superconducting grains.

[1] Université Paris-Saclay, CNRS/IN2P3, CSNSM, 91405 Orsay, France. [2] Unité Mixte de Physique, CNRS, Thales, Université Paris-Saclay, 91767 Palaiseau, France. [3] Departamento de Física—CIOyN, Universidad de Murcia, 30100 Murcia, Spain. [4] Université Paris-Saclay, CNRS/IN2P3, IJCLab, 91405 Orsay, France. ✉email: moo@um.es; claire.marrache@ijclab.in2p3.fr

T he recent interest in low-dimensional systems primarily stems from the many exotic electronic phases they exhibit, due to their extreme sensitivity to any emerging order[1–3]. The superconductor−insulator transition (SIT) in two-dimensional disordered materials is one such famous example. There, super-conductivity competes with localization and Coulomb interactions to give rise to unusual electronic phases[4–9].

Despite several decades of investigation of the SIT, the nature of the insulator is still a subject of debate. Some argue that Coulomb interactions and disorder prevail so that the insulator is fermionic[10,11], whereas others claim that localized Cooper pairs exist even if global superconducting coherence is suppressed[12,13]. Moreover, there is a controversy on whether the system is electronically homogeneous, granular, or fractal[6,14]. Especially intriguing are the reports of very strong insulating behaviors in the immediate vicinity of the SIT, with activated or even overactivated temperature dependence of the resistivity at the lowest experimentally accessible temperatures[15–24]. These findings have prompted fierce debate as to their possible interpretation.

Arrhenius, or activated, behavior is found in the electronic transport of many insulators. In this case, the temperature dependence of the resistance is of the form:

$$R = R_0 \, \exp\left(\frac{T_0}{T}\right), \qquad (1)$$

where $T_0$ is the activation temperature. It is usually associated either with a band gap or with nearest neighbor hopping. In disordered materials, the insulating behavior originates from charge carriers being spatially localized, and activated behavior usually takes place at relatively high temperatures[25]. At low temperatures, electrons have to compromise between tunneling to close neighbors at the price of an energy mismatch, or traveling further where the hopping energy difference may be smaller. This process is known as variable-range hopping (VRH)[26] and results in a temperature dependence of the resistance of the form:

$$R = R_0 \, \exp\left(\frac{T_0}{T}\right)^\alpha. \qquad (2)$$

For non-interacting systems, Mott[27] predicted an exponent $\alpha = D/(D+1)$, with $D$ the system dimensionality. Efros and Shklovskii[28] extended the argument to Coulomb interacting systems and obtained an exponent of 1/2, independent of dimensionality. In both cases, the temperature dependence is subactivated, i.e. $\alpha < 1$. Experimentally however, in systems close to the SIT, activated[29–31] and, in some cases, overactivated dependencies are found at very low temperatures[15–22]. This unfamiliar behavior calls for investigation and is the subject of the present work.

Here, we investigate, both experimentally and numerically, transport properties of quasi-2D systems on the insulating side of the SIT. We have measured over 80 insulating $Nb_xSi_{1-x}$ films down to 7 mK to span several orders of magnitude in activation temperatures. We perform Monte Carlo simulations of VRH conductivity on quasi-2D systems, so as to reproduce the experimental situation. We analyze at which conditions activated behavior is present at very low temperatures and investigate the different scenarii leading to overactivated transport. We compare our experimental results with the outcome of our simulations to extract the main physics of conduction mechanisms in the insulating state of the SIT. Our main conclusion is that, close to the SIT, appearance of Cooper pairing enhances the activation temperature at low temperature.

## Results and discussion

**Experimental system.** Thin metal alloy films constitute model systems to study the zero-magnetic field SIT. In these compounds, the SIT can be driven either by a reduction of the sample thickness or by a variation in stoichiometry which directly affects the amount of disorder in the system. In amorphous $Nb_xSi_{1-x}$ (a-NbSi) which we consider, thermal treatment is yet another parameter that enables us to very finely and progressively tune a single sample disorder[32,33]. The combination of these experimental tuning parameters allows us to study the effects of thickness and disorder independently.

We have considered over 80 insulating a-NbSi thin films with compositions ($x = 9\%$ and 13.5%), thicknesses ($d \in [20, 250]$ Å) and heat treatments ($\theta_{ht} \in [70, 160]°$C) that allowed us to investigate localized regimes with activation temperatures spanning over four decades.

The as-deposited sample conductivity has been studied from room temperature down to 7 mK (see 'Methods'). The corresponding temperature dependence of the samples sheet resistance is shown in Fig. 1 (panel (a) $x = 13.5\%$ batch, panel (c) $x = 9\%$ batch). The films considered in this work are all insulators and, as expected, a thickness reduction drives the system deeper into the insulating regime. All $x = 13.5\%$ samples have been progressively annealed from 70 to 160 °C, to slightly change the amount of disorder in the films. Panels (d)−(f) of Fig. 1 show the effect of successive heat treatments on a single sample: in our case, the higher the heat treatment temperature $\theta_{ht}$, the more insulating the film becomes[32].

For all films and all heat treatments, the low temperature sheet resistance follows an Arrhenius-type law as given by Eq. (1). This is particularly visible in Fig. 1 (panels (b) and (c)). The $x = 9\%$ samples show a single activated behavior, while the $x = 13.5\%$ samples deviate from this law at the lowest temperatures. Previous works have reported such a behavior[15,19,20,22–24,34]. It has often been referred to as overactivation and associated with the presence of superconductivity in the system[15,19,20,23,24]. However, this intriguing feature has mostly been observed under magnetic field[19,20,23,24,34] or pressure[22], and has not been systematically addressed experimentally. In particular, the role disorder plays in the emergence of this behavior has not yet been settled, neither has the nature of the excitations that are dominant for low frequency transport been determined. As can be seen from Fig. 1g, the resistance actually undergoes a crossover from an activated regime to an overactivated regime at very low temperatures, with a slightly larger characteristic energy $T_0'$. A quantitative understanding of both regimes is one of the main goals of this work.

**Activated behavior.** Let us first study the activated regime and derive the expression for its characteristic temperature. Close to the SIT, one expects thin films to have very large localization lengths $\xi_{loc}$ and therefore very high dielectric constants $\kappa$. As a consequence, the electric field lines will tend to stay within the film, resulting in an approximately logarithmic effective interaction[26,35]:

$$V(r) \approx \frac{e^2}{2\pi\epsilon_0\kappa d}\ln\frac{r_{max}}{r}, \qquad (3)$$

for distances $r$ between $r_{min}$ and $r_{max}$. $r_{min}$ corresponds to a typical grain radius in granular materials and to $\xi_{loc}$ otherwise. $r_{max}$ usually corresponds to the electrostatic screening length $\kappa d$. Due to these logarithmically long-range Coulomb interactions, the insertion of a charge into the system produces electric

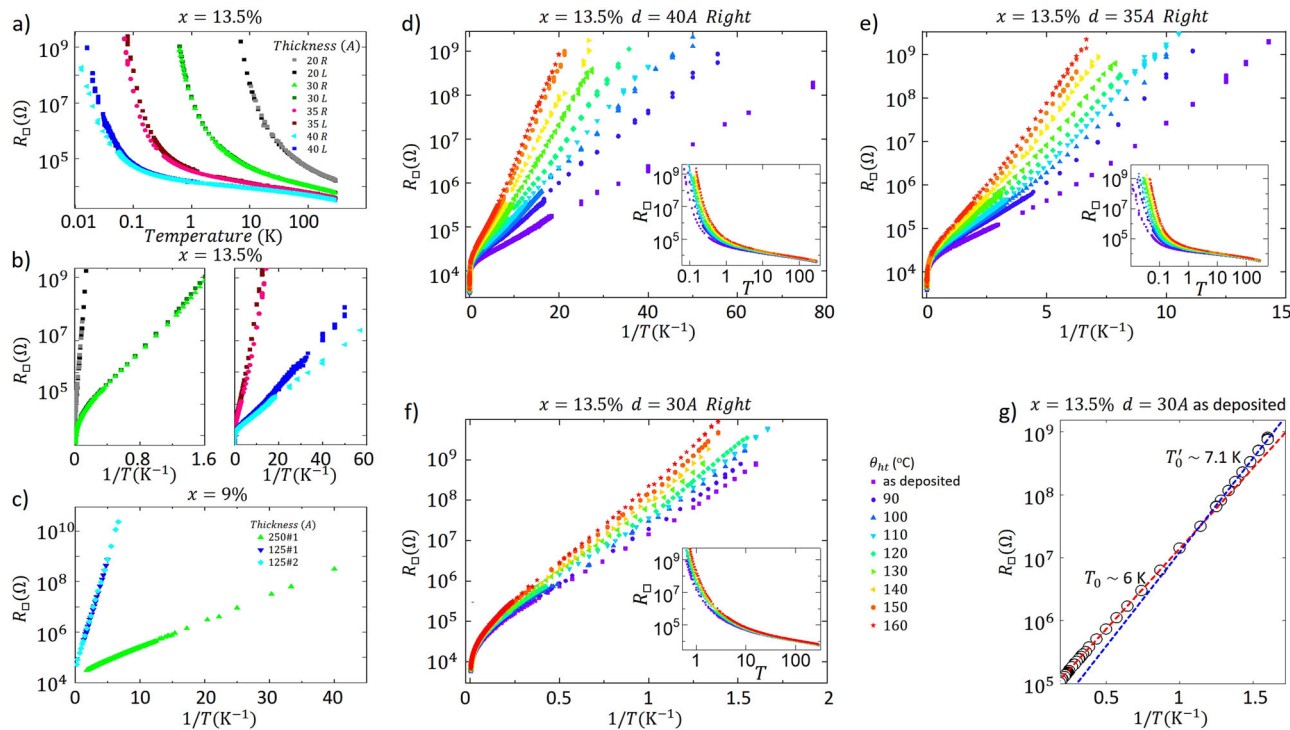

**Fig. 1 Low temperature resistance measurements. a** Temperature dependence of the sheet resistance of $Nb_xSi_{1-x}$ for the $x = 13.5\%$ as-deposited batch. **b** Sheet resistance as a function of $1/T$ on a semilogarithmic scale to highlight the activated behavior for the $x = 13.5\%$ samples. R and L correspond to the right and left sides of the same sample (see 'Methods'). **c** Sheet resistance as a function of $1/T$ on a semilogarithmic scale for the as-deposited samples. **d−f** Effect of successive heat treatments on the sheet resistance of the 13.5% 40 Å (**d**), 35 Å (**e**) and 30 Å (**f**) samples. The insets show the $R_□(T)$, while the main panels display the same data as a function of $1/T$. **g** At low temperature, at about 0.9 K for the 13.5% 30 Å as-deposited film, there is a crossover between two activated regimes. Circles represent the data and dashed lines correspond to linear fits with the corresponding activation temperatures ($T_0$ and $T'$) represented.

fields over a large region, resulting in a charging energy:

$$E_0 = \frac{1}{2}\epsilon_0\kappa \int dr |E(r)|^2 \approx \frac{e^2}{4\pi\epsilon_0\kappa d}\ln\frac{r_{max}}{r_{min}} \approx kT_0, \quad (4)$$

where $k$ is Boltzmann constant, and the dielectric constant $\kappa$ is given by[36]:

$$\kappa = \kappa_0 + 4\pi\beta_2\frac{e^2}{a}N(E_F)\xi_{loc}^2, \quad (5)$$

$\kappa_0$ being the host dielectric constant, $\beta_2 \approx 3.2$, $a$ the typical interatomic distance and $N(E_F)$ the 2D density of states at the Fermi level. This charging energy opens a gap in the single-particle density of states. In the presence of disorder, extending Efros and Shklovskii's argument for the Coulomb gap[37] to our logarithmic interaction, Eq. (3), one obtains an exponential density of states around the Fermi level with a characteristic energy $kT_0$ proportional to this charging energy[37,38]. Extending Mott's VRH argument to this density of states results in an activated conductivity with logarithmic corrections[26,39]. As a consequence, a low temperature activated behavior can be the result of either transport dominated by charging energies or VRH in quasi-2D materials with high dielectric constants.

To study these two scenarii, we model the electrical transport in these disordered films by a 2D random capacitor network schematized in Fig. 2a (see 'Methods' and ref. [40]). Grains are interconnected by random capacitors with average capacitance $C$ and connected to the gate by a capacitance $C_0$. Conduction is by hops of quantized charges between grains. By a proper choice of parameters, the model can generate regimes either controlled by charging energies, or dominated

by logarithmic Coulomb interactions between charges. A gate capacitance $C_0 \ll C$ results in a regime dominated by long-range logarithmic Coulomb interactions with a screening length, $\kappa d$, given by $\sqrt{C/C_0}$ [41]. If $C \lesssim C_0$ the only relevant energies involved are the grain charging energies. The hopping conductivity in the system is calculated using a kinetic Monte Carlo method[42,43].

The results of the simulations are shown in Fig. 2b, where we represent the resistance for three values of the gate capacitance. Solid lines correspond to clean samples and dashed lines to disordered samples where 5% of the nodes have fixed charges ±1, at random, not contributing to the current and creating a random onsite potential. We observe a roughly activated behavior at low $T$ in all cases. The activation energy increases as $C_0$ decreases and the screening length increases.

Our results show that disorder produces a decrease of the activation energy. In our simulations, disorder is introduced by placing fixed charges at random. Carriers will move through regions where the long-range contribution of the interaction is screened by other charges. The presence of such charged impurities will reduce $r_{max}$ roughly to the inter-impurity distance. In real situations, any disorder will induce the presence of charges, so that $r_{max}$ will be of the order of the distance between these charges. All samples then end up with very similar activation temperatures, of the order of $kT_0 = E_0$.

Logarithmic interactions in disordered systems imply the existence of charging energies, and so effective electronic granularity, irrespective of the film morphology[44]. We can think of our samples as formed by effective grains with a charging energy given by Eq. (4).

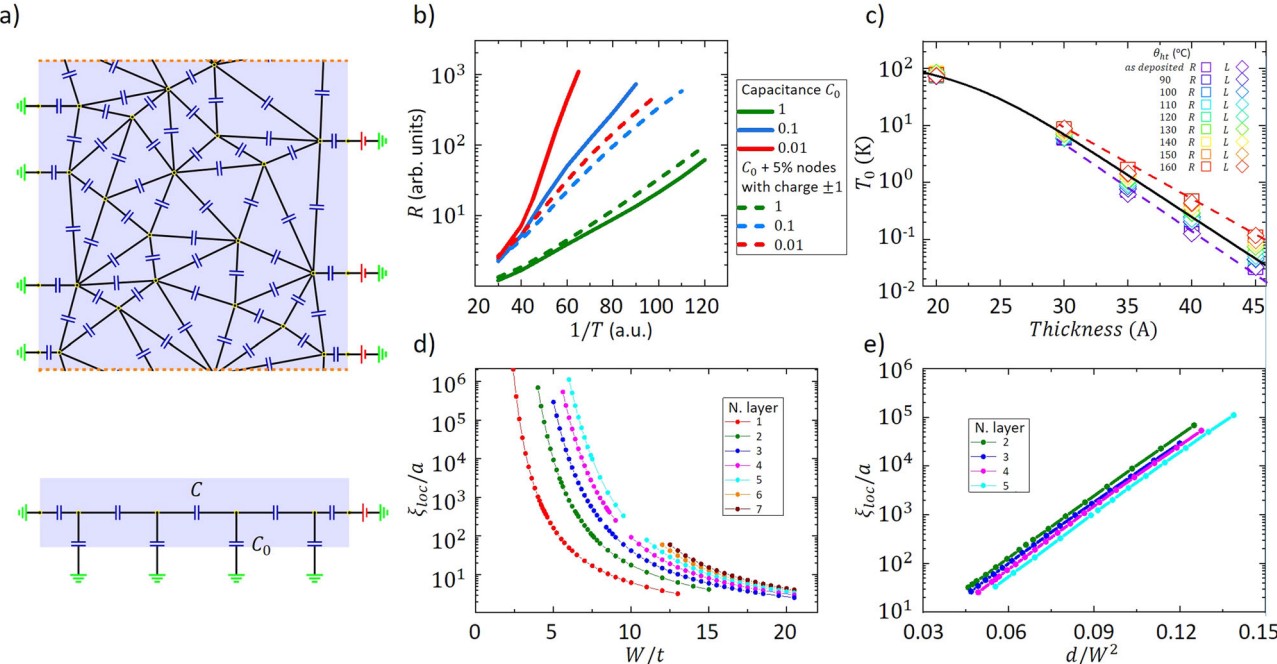

**Fig. 2 Numerical simulations of the localization length and comparison with experimental data. a** A sketch of the model used for numerical simulations. The electrode connected to the left edge is grounded, the electrode at the right edge has a potential $V$. The bottom panel shows a lateral view of the model where the capacitors $C_0$ connecting the system to the gate can be seen. Grains are interconnected by random capacitors with average capacitance $C$. **b** Numerical results of the resistance as a function of inverse temperature. Each color corresponds to a different value of $C_0$. The solid lines correspond to the clean case whereas the dashed lines correspond to the case where disorder is added (5% of the nodes have fixed charges ± 1 at random). **c** Characteristic temperature $T_0$ as a function of the sample thickness (in Å) for different heat treatments ($x = 13.5\%$ sample). The solid line corresponds to Eqs. (4)-(6) with $\kappa_0 \simeq 250$. For each thickness, the disorder level was averaged over the different heat treatments. Each dashed line corresponds to the exponential thickness dependence for a given heat treatment (as-deposited and $\theta_{ht} = 160\,°C$). **d** Localization length on a logarithmic scale as a function of the disorder level $W$ (relative to the kinetic energy $t$) for thicknesses ranging from 1 to 7 layers. **e** Exponential dependence of $\xi_{loc}$ with $d/W^2$, where $d$ is the thickness.

**Exponential dependence of the localization length with thickness**. Let us come back to the experimental situation. In Fig. 2c, the activation temperature $T_0$ is plotted as a function of sample thickness for $x = 13.5\%$ and all heat treatment temperatures. The first striking feature of this plot is that $T_0$ varies over almost four orders of magnitude. Except for the thinnest sample ($d = 20\,\text{Å}$), $T_0$ presents an exponential dependence on sample thickness, such that $T_0 \propto e^{-\zeta d}$, with $\zeta$ that depends on the heat treatment temperature, i.e. the degree of disorder.

According to Eqs. (4) and (5), when the localization length is large, $T_0 \sim (\kappa d)^{-1} \sim \xi_{loc}^{-2} d^{-1}$. To explain the exponential dependence of $T_0$ on thickness observed experimentally, $\xi_{loc}$ must increase exponentially with thickness. To investigate this dependence, we have calculated $\xi_{loc}$ for square samples of finite thickness in an Anderson model, through a one-parameter scaling analysis of their conductance. The main results of these simulations are presented in Fig. 2d where we plot $\xi_{loc}/a$ as a function of $W/t$ ($W$ is the disorder level and $t$ the kinetic energy of the electrons) for thicknesses ranging from $d = 1$ to 7 layers.

For small disorder levels, one can appreciate the strong dependence of $\xi_{loc}$ on both $W$ and $d$. More quantitatively, our numerical simulations establish that in this regime the localization length is of the form (Fig. 2e):

$$\xi_{loc} \approx A \exp\left\{\frac{\eta d}{W^2}\right\} \qquad (6)$$

with $A$ and $\eta$ approximately constants, but non-universal. The small shift between data for different thicknesses can be taken into account with a thickness-dependent prefactor.

Although the exponential dependence of the localization length with thickness has been observed in many systems, it is often attributed to a thickness-induced change in the amount of the disorder $W$. We are here able to distinguish the quite distinct effects of both parameters. Equation (6) is in line with the self-consistent theory of Anderson localization[45,46], that predicts this exponential dependence of the localization length $\xi$ on disorder $W$ for 2D systems. It is also in agreement with analytical self-consistent results in weakly localized 2D systems[47], with numerical simulations performed on quasi-1D systems[48], as well as with the exponential dependence of the localization length with thickness was observed in Mo-C films[49] in VRH regimes.

The activation temperature therefore exponentially decreases with thickness as shown by the straight lines in Fig. 2c. Moreover, we expect each $\theta_{ht}$ to correspond to a single value of $W$, so that the disorder-induced spread in $T_0$ should linearly depend on the thickness, which is experimentally the case in our films. For small thicknesses, the contribution of the host dielectric constant $\kappa_0$ becomes non-negligible, causing the bending downwards of the solid curve in Fig. 2c for $d \lesssim 30\,\text{Å}$ and the values of $T_0$ are smaller than the exponential prediction.

**Overactivation**. We will now turn to the overactivated behavior, i.e. the increase in the activation energy at the lowest temperatures, in the immediate vicinity of the SIT. Let us stress that we here concentrate on the zero-magnetic field situation.

Far from the SIT, the thinnest ($d = 20\,\text{Å}$) samples do not exhibit any overactivated regime. However, samples closer to the SIT do. To appreciate the systematicity of the overactivated behavior, we scaled the high temperature activated regime for the

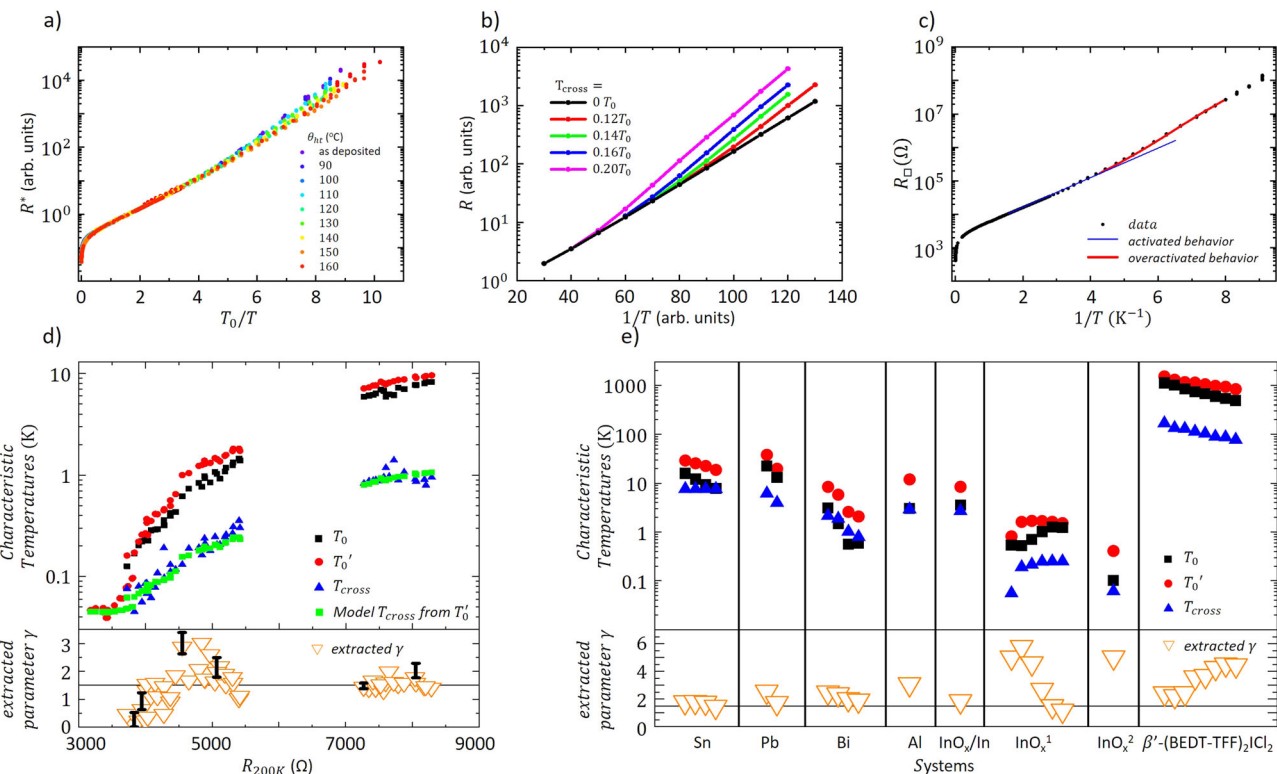

**Fig. 3 Characterization of the activated and overactivated regimes. a** Scaled resistance as a function of $T_0/T$ (with $T_0$ the activation temperature and $T$ the temperature) so that the activated regime overlaps for all disorder levels (tuned by the successive heat treatments) for the 35-Å-thick 13.5% sample. The unscaled data are shown in Fig. 1e. **b** Simulation results for the resistance as a function of $1/T$, on a semilogarithmic scale. The black line corresponds to activated behavior, with no superconductivity, and the other curves to overactivation due to superconductivity for several values of the ratio $T_{cross}/T_0$, with $T_{cross}$ the crossover temperature between the activated and overactivated regimes. The overactivated regime is characterized by an activation temperature $T'$ as shown Fig. 1g. **c** Resistance as a function of the inverse temperature, $1/T$, on a semilogarithmic scale, for the 13.5% 35-Å-thick sample (heat treatment of 110 °C, black dots). The continuous blue curve is our simulation without superconductivity effects, while the red curve is with grain superconductivity for the case $T_{cross}/T_0 = 0.14$. **d** Characteristic temperatures $T_0$, $T_0'$ and $T_{cross}$ for all measured samples. The bottom panel shows the value of $\gamma = (T' - T_0)/T_{cross}$ as given by Eq. (9). The black line then corresponds to the average $\gamma_{exp} = 1.5$. The green squares correspond to $T_{cross}$ given by Eq. (10): $T_{cross} \simeq 1.8/d\,[\text{Å}] + 0.105 \times T_0'$. The black error bars are typical error bars on $\gamma_{exp}$ at the considered disorder: far from the SIT, the error bars are typically smaller than closer to the SIT, where the two activated regimes are difficult to precisely define. **e** Characteristic temperatures $T_0$, $T_0'$ and $T_{cross}$ of the activated regimes in several systems (re-analysis of published data). The bottom panel gives the value of $\gamma$, determined by Eq. (9). The black line corresponds to $\gamma_{exp} = 1.5$ obtained from our results on a-NbSi. The experimental data are from ref. [15] for Sn and Pb[16], for Bi[50], for Al[17], for InOx/In (Shahar, D. Transport measurements of InOx films, private communication), and ref. [23] for InOx under magnetic fields (respectively InOx[1] for fields ranging from 1 to 7 T and InOx[2] taken at 0.75 T), and ref. [22] for β'-(BEDT-TFF)$_2$ICl$_2$ under pressures ranging from 0 to 1.8 GPa.

13.5% 35-Å-thick sample (Fig. 3a) at different heat treatment temperatures (i.e. disorder). By doing so, the overactivated regimes cannot be made to scale. Moreover, the ratio of the overactivated characteristic temperature $T_0'$ over the activated one $T_0$ decreases with the disorder level.

We have extracted $T_0'$, whenever an overactivated regime was observed, by fitting the low temperature data to Eq. (1). The transition temperatures $T_{cross}$ between the two activated regimes were also extracted from the crossing point between the two fits. The corresponding results are plotted in Fig. 3d as a function of the disorder level, quantified by the high temperature (200 K) resistance $R_{200\,K}$. We note that $T_0$, $T_0'$ and $T_{cross}$ vary over almost three orders of magnitude. Moreover, $T_{cross}$ gets closer to $T_0$ as we approach the SIT, at about $R_{200\,K} \simeq 3450\,\Omega$. Our aim is now to understand the relation between these characteristic temperatures since they shine light on the conduction mechanisms.

Since the overactivated regime is only found close to the SIT, it is reasonable to assume it is related to superconductivity. Let us examine what will happen if, due to the proximity to the SIT, superconductivity sets in locally in some (effective) grains. In

this case, Cooper pairs will be phase coherent on a short range, but there will not be any global phase coherence. There are then two different possibilities. The first one is that conduction is still dominated by one-electron processes. Then, when a single electron enters a superconducting grain, it has to pay an extra energy penalty associated with the superconducting gap. The second possibility is that conduction between superconducting grains is ensured by tunneling of Cooper pairs. In this case the energy penalty for a pair entering a superconducting grain is four times the charging energy, since the grain becomes doubly charged. In both cases, the crossover temperature to the overactivated regime must be of the order of the pairing energy. In our experimental case, $T_{cross}$ is much smaller than the activation energy ($kT_{cross} \ll kT_0 \sim E_0$), i.e. the pairing energy is much smaller than the charging energy, so that conduction by single electrons is more likely to be the main conduction mechanism.

We have performed Monte Carlo simulations of the conductivity on our random capacitor model to quantify the increase in activation energy when the grains become superconducting.

We consider the model with only charging energies and no long-range interactions, since, as we have seen, both cases produce similar results, and the higher computational efficiency of the former is convenient in this highly demanding calculation.

The grains become superconducting typically below $T_{cross}$, so that the energy penalty for an electron to enter a grain becomes:

$$E' = E_0 + \Delta(T) \, , \tag{7}$$

where $\Delta$ is the superconducting gap. Assuming $\Delta/kT_{cross} = 2$ (Chapelier, M. STM measurements of $Nb_xSi_{1-x}$ films, private communication 2011), we have:

$$E' = E_0 + 2kT_{cross}\sqrt{\frac{T_{cross} - T}{T_{cross}}} \, . \tag{8}$$

Let us emphasize that the random nature of our capacitor model naturally takes into account the fluctuations of the gap amplitude that are bound to exist in such disordered systems[7].

The results of the simulations are shown in Fig. 3b, where we plot $R$ versus $1/T$ for several values of $T_{cross}$ (in units of $T_0$). The black curve corresponds to the non-superconducting case (pure activated behavior) and the rest to different values of $T_{cross}/T_0$, chosen to reproduce the experimental situation. At temperatures below $T_{cross}$, we obtain a new roughly activated regime, as experimentally observed. As expected, the transition temperature between both regimes coincides with $T_{cross}$.

In Fig. 3c we compare our numerical simulations for $T_{cross} = 0.14T_0$ with the experimental data for the 35 Å sample ($\theta_{ht} = 110\,°C$) by scaling the theoretical curve so that both coincide in the activated regime. The overactivated behavior is fairly well reproduced by our simulations.

Extracting characteristic temperatures from our numerical results in the same way as from the experimental data, we have established the relation:

$$T' - T_0 = \gamma T_{cross} \tag{9}$$

and found $\gamma_{num} = 1.8$. Independently for each experimental point, we derived the value of $\gamma_{exp}$ given by Eq. (9) and represent it as an orange triangle (bottom panel of Fig. 3d). On average, we find $\gamma_{exp} = 1.5$ (black line). The agreement with theoretical predictions is fairly good, clearly indicating that the overactivated regime is a consequence of electrons having to pay the superconducting gap penalty, proportional to $T_{cross}$. Note that, close to the SIT, uncertainties on $\gamma_{exp}$ are large, since the activated regime expands on a very narrow temperature range.

We have also reanalyzed the stronger-than-activated behaviors found in the literature (Shahar, D. Transport measurements of InOx films, private communication 2020)[15–17,22,50]—although it may not have been originally analyzed as such—and compared the experimental data to our prediction. In Fig. 3e, we represent $\gamma$ obtained from Eq. (9) for each system (orange triangles). The value of $\gamma$ is fairly constant for all systems, and of the order of unity. This is remarkable given the variety of systems considered, and the three orders of magnitude over which $T_0$ extends. We therefore believe that our theoretical scenario could be an alternative explanation for the stronger-than-activated behaviors reported in the literature[23,24], with the caveat that we do not here consider the effect of magnetic field or pressure.

Let us note that the theoretical prediction depends both on the BCS ratio linking $T_{cross}$ and $\Delta$, which we took to be 2 following STM data for a-NbSi (Chapelier, M. STM measurements of $Nb_xSi_{1-x}$ films, private communication), and on the proportion of superconducting grains. If, instead of having all grains superconducting, only a fraction $p$ of them are, $T' - T_0$ should be proportional to $p$. From the agreement between the

simulations and the experiment, we can conclude that a large fraction of the grains become superconducting.

We can explain the dependance of $T_{cross}$ on disorder (Fig. 3d) by assuming that the superconducting gap is of the form

$$\Delta = \frac{\Delta^{3D}\xi_{SC}}{d}\left(1 + \frac{\xi_{SC}^2}{\xi_{loc}^2}\right) \sim kT_{cross}, \tag{10}$$

where $\xi_{SC}$ is the superconducting coherence length in the material and $\Delta^{3D}$ is the bulk superconducting gap. This equation is an interpolation between the gap for a uniform system of finite thickness[51–56] and the situation of small superconducting grains. The first limit occurs close to the SIT when $\xi_{loc}$ is large. Then, $\Delta \simeq \frac{\Delta^{3D}\xi_{SC}}{d}$ is approximately constant. In the second limit, farther from the SIT, the equation for the gap becomes similar to the proposal in ref. [57]: it increases as the inverse of the volume over which Cooper pairs are forced to be confined.

Far from the SIT, since the fastest changing parameter in $T_0$ is $\xi_{loc}^{-2}$ (Eqs. (4) and (5)), Eq. (10) implies that $T_{cross}$ is of the form $a/d + bT_0$. Green squares in Fig. 3d are a fit of $T_{cross}$ by this expression. The agreement with experimental points (blue) is relatively good, although, close to the SIT, the temperature range over which $T_0$ is observed is small and the uncertainty large. Close to the SIT, for our 13.5% 45-Å-thick samples, the activation energy is almost constant because it is dominated by the superconducting gap, larger than the charging energy, and approximately constant and equal to $(\xi_{SC}/d)\Delta^{3D}$. In this region, we observe only one activated regime and, since there is no crossover, the green squares correspond to a prediction of the superconducting temperature of the grains.

Strictly speaking, $\xi_{loc}$ in Eq. (10) should be the Cooper pair localization length, that, due to interactions, may be larger than the one particle localization length, but this difference is small in the regime considered and we have assumed that both were equal.

We have shown that the activated transport behavior observed in thin films close to the SIT is due to transport dominated by charging energies. In homogeneous systems, the electronic granularity is a consequence of a diverging localization length. In a-NbSi, we have established that, for a fixed amount of disorder, $\xi_{loc}$ depends exponentially on the film thickness. The overactivated regime observed close to the SIT is a crossover to a regime governed both by the charging energy and the superconducting gap. This is surprising in a material where no sign of bosonic insulator was observed until now. Our results indicate that the superconducting gap depends critically on the grains size. The overall conclusion is that, in the insulating regime close to the SIT, localized Cooper pairs exist but electronic transport is still dominated by single electrons.

## Methods

**Experimental details**. The a-NbSi films have been grown at room temperature by e-beam co-deposition of Niobium and Silicon under ultra-high vacuum (the chamber pressure during the deposition was typically of a few $10^{-8}$ mbar). The film composition was fixed by the respective evaporation rates of Nb and Si (both of the order of $1\,Å\,s^{-1}$) and monitored in situ by a set of dedicated piezoelectric quartz crystals. The sample thickness was determined by the duration of the deposition. Both parameters have been checked ex situ by Rutherford Back-scattering Spectroscopy (RBS).

The samples have been deposited onto sapphire substrates coated with a 25-nm-thick SiO underlayer designed to smooth the substrate surface. They were also protected from oxidation by a 25-nm-thick SiO overlayer. a-NbSi films of similar compositions and thicknesses have been measured to be continuous, amorphous and homogeneous at least down to the thickness 2.5 nm[32].

The transport characteristics of a-NbSi thin films are mainly determined by their composition $x$ and their thickness $d$. However, an additional thermal treatment can also microscopically modify the system disorder while keeping $x$ and $d$ constant. a-NbSi becomes more insulating as the heat treatment temperature increases without any change in the sample morphology[32]. Thermal treatments

and the film composition have an analogous effect on the disorder level: films of similar sheet resistance $R_\square$ have the same transport characteristics. On the other hand, we have shown that the effect of the thickness is distinct[32]. In the present case, we have considered as-deposited films which parameters are listed in Table 1. The composition $x$ has been chosen so that the samples are close to the SIT ($x = 13.5\%$) or further within the insulating regime ($x = 9\%$). The sample thickness has been varied between $d = 20$ and 50 Å for $x = 13.5\%$. For this stoichiometry, the critical thickness at which the system undergoes a thickness-tuned SIT is about 140 Å [58]. a-Nb$_9$Si$_{91}$ is not superconducting, even for bulk samples[59,60], and in this work, we considered thicknesses of 125 and 250 Å. Each batch having a constant composition is considered to have the same disorder level ($W$ constant). We can therefore directly evaluate the effect of the thickness within each batch.

The samples resistances have been measured in a dilution refrigerator with a base temperature of 7 mK. In the case of the $x = 13.5\%$ batch, two regions of each sample, labeled left and right, could be probed independently. We used standard low noise transport measurement techniques to ensure the samples were measured without electrical heating of the electronic bath. In other words, we made sure via appropriate filtering that the sample electron temperature was the same as its phononic temperature. Moreover, we have checked that the applied bias was sufficiently low for the resistance measurement to be in the ohmic regime.

**Capacitor model**. To calculate the hopping conductance of quasi-2D high dielectric constant disordered systems, we consider a 2D random capacitor model[40] in order to get a consistent set of energies. Sites (nodes) are randomly distributed and the corresponding junction network without crossings can be constructed using a Delaunay triangularization algorithm[61]. Capacitors with randomly chosen capacitances were placed at the links between adjacent nodes, as shown in Fig. 2a. The capacitances $C_{i,j}$ assume random values drawn from the distribution $C_{i,j} = Ce^\varphi$, where $\varphi \in [-B/2, B/2]$. We have chosen $C = 1$ (in units of $\epsilon_0 a$, where $a$ is the lattice constant) and $B = 2$. The left bank is connected to the ground, while the right bank is at the potential $V$, and there are periodic boundary conditions in the lateral direction. In order to take into account the leakage of field lines to outside the 2D system due to the finite value of $\kappa$, we introduce capacitances to the ground, $C_0$, as shown in the bottom panel of Fig. 2a.

The plates of each capacitor carry opposite charges and the total charge on each site $Q_i$ is the sum of the charges on the plates of the capacitors connected to it, $Q_i = \sum_j q_{i,j}$. Here, $q_{i,j}$ is the charge on the plates of the capacitor connecting nodes $i$ and $j$ and satisfies

$$q_{i,j} = \sum_j C_{i,j}(V_i - V_j) . \tag{11}$$

We construct a vector $\mathbf{Q}$, whose components are the charges of the nodes in the sample, and a vector $\mathbf{V}$, whose components are the node potentials. Both vectors are related by $\mathbf{Q} = \mathbb{C}\mathbf{V}$ where the capacitance matrix $\mathbb{C}$ has components equal to

$$[\mathbb{C}]_{i,j} = \left(\sum_{k \neq i} C_{i,k}\right)\delta_{i,j} - C_{i,j} . \tag{12}$$

The total electrostatic energy of this system can be obtained in a compact form through the inverse of the capacitance matrix[40]

$$H = \frac{1}{2}\mathbf{Q}\mathbb{C}^{-1}\mathbf{Q}^T . \tag{13}$$

The matrix $\mathbb{C}^{-1}$ plays the role of an interaction matrix. In the continuous limit, the average interaction between two charges separated by a distance $r$ is

proportional to the modified Bessel function $K_0(r/\Lambda)$, where the screening length is $\Lambda = r_0\sqrt{C/C_0}$ [41]. In the limit of small $r$ ($r_0 \ll r \ll \Lambda$) we have $K_0(x) \approx -\ln(x/2)$, and the effective interaction is given by Eq. (3) provided that we identify the screening length with $\Lambda = \kappa d$, and choose $C = \epsilon_0\kappa d$ and $C_0 = \epsilon_0 r_0^2/(\kappa d)$. In the simulation of hopping transport, $r_0$ is our unit of distance, which for random nodes is defined by $r_0 = L/N^{1/2}$, $N$ being the number of nodes, $L$ the size of the system, and $e^2/(r_0\epsilon_0)$ is our unit of energy. The charging energy, i.e., the energy cost for putting a unit charge at a given site $i$, is equal $\mathbb{C}_{i,i}^{-1}/2$.

Carriers hop from site to site transferring a quantized unity charge. They can hop over many sites with a probability that exponentially decays with distance, thus modeling charge transfer in experimental systems mediated by the cotunneling mechanism. The transition rate between sites $i$ and $j$ can be expressed as:

$$\Gamma_{i,j} = \tau_0^{-1}e^{-2r_{i,j}/\xi_{\text{loc}}}e^{-\Delta_{i,j}/kT}, \tag{14}$$

where $\tau_0^{-1}$ is the phonon frequency, $r_{i,j}$ the hopping distance, $\xi_{\text{loc}}$ the localization length and $\Delta_{i,j}$ the transition energy, given by our capacitor model. To simulate hopping conductivity in the system of interacting electrons, we employ a kinetic Monte Carlo method[42,43]. The allowed node charges are 0 and ±1. At each Monte Carlo step, the algorithm chooses a pair of sites $(i,j)$ with the probability proportional to $\exp(-2r_{ij}/\xi)$, ref. [42]. Doing so, the time step associated with a hop attempt per site is $\tau_0/\sum_{ij}\exp(-2r_{ij}/\xi)$, where $\tau_0$ is the inverse phonon frequency. The algorithm first checks whether the transfer of a unit charge from site $i$ to $j$ is compatible with the allowed node charges. Then it calculates the transition energy

$$\Delta_{i,j} = V_j - V_i - \mathbb{C}_{i,j}^{-1} + \frac{1}{2}\left[\mathbb{C}_{i,i}^{-1} + \mathbb{C}_{j,j}^{-1}\right] , \tag{15}$$

where $V_i = \sum_j \mathbb{C}_{i,j}^{-1}Q_j$ is the potential at $i$, and the hop is performed when $\Delta_{i,j}$ is negative or with probability $\exp(-\Delta_{i,j}/T)$ otherwise. All site potentials $V_i$ are recalculated after every successful transition. The last term in Eq. (15) is the charging energy of the nodes involved. The electric current is generated by the potential difference $V_{\text{lead}}$ between the leads at the opposite sides of the sample, which is reflected in the site potentials by extending the $\mathbf{Q}$ vector to include nodes $j$ in the lead connected to nodes $i$ in the sample and associating a charge $-C_{i,j}V_{\text{lead}}$ with each of them[40].

The algorithm starts from an initial random charge configuration and follows the dynamics at a given temperature. Once the system is in a stationary situation, the conductivity of each sample is calculated from the number of electrons crossing to one of the leads. The number of Monte Carlo steps performed in this calculation drastically increases with decreasing $T$, and it is determined by the condition that the net charge crossing one of the leads is on the order of 1000. The number of samples considered is 100. Finally, we averaged $\ln \sigma$ over the set of samples (an ensemble averaging).

The main results of the simulations were shown in Fig. 2b. From them, we concluded that, in the presence of disorder, the activation energy is fairly independent of the screening length or, equivalently, of the ratio $C_0/C$. To simulate the overactivated regime, we have to use the most efficient numerical procedure in order to reach very low temperatures. To this end, in Fig. 4 we compare the resistance for the exact interacting potential in the $C_0 = 1$ case (thick curves) with the results including only the charging energy (thin curves). Again, thick lines correspond to the case without site disorder, while thin lines reflect samples with 5% of the nodes having fixed random charges. We see that the activation energies are similar and conclude that a model with charging energies only, without the logarithmic contribution, is adequate to simulate conductivity in disordered samples.

For the simulations of the overactivated regime, the random capacitor network takes into account the spatial variation of the superconducting gap amplitude[7]. Indeed we have taken a gap varying by about 40%, thus reflecting the fluctuations in $\Delta$ that have been measured in similar films (Chapelier, M. STM measurements

**Table 1 Characteristics of the different a-NbSi samples: composition $x$, thickness $d$, low temperature sheet resistance $R_{4K}$ evaluated at 4.2 K except for the 20-Å-thick sample for which they have been measured at 5 K.**

| Name | $x$ (%) | $d$ (Å) | $R_{4K}$ |
|---|---|---|---|
| CKSAS43 $\alpha$ left | 13.5% | 20 | 111 GΩ |
| CKSAS43 $\alpha$ right | 13.5% | 20 | n.d. |
| CKSAS61 $\alpha$ left | 13.5% | 30 | 162 kΩ |
| CKSAS61 $\alpha$ right | 13.5% | 30 | 151 kΩ |
| CKSAS61 $\beta$ left | 13.5% | 35 | 18.8 kΩ |
| CKSAS61 $\beta$ right | 13.5% | 35 | 17.8 kΩ |
| CKSAS61 $\gamma$ left | 13.5% | 40 | 10.8 kΩ |
| CKSAS61 $\gamma$ right | 13.5% | 40 | 10.5 kΩ |
| CKSAS61 $\delta$ left | 13.5% | 45 | 7.73 kΩ |
| CKSAS61 $\delta$ right | 13.5% | 45 | 7.50 kΩ |
| CK8 $\gamma$ | 9% | 125 | 53.3 kΩ |
| CK9 $\gamma$ | 9% | 125 | 52.6 kΩ |
| CK9 $\beta$ | 9% | 250 | 14.1 kΩ |

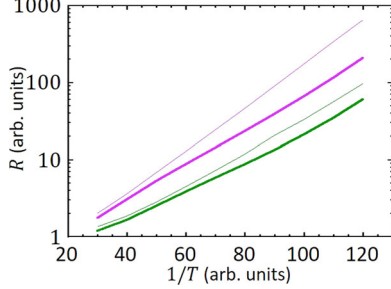

**Fig. 4 Numerical results.** Resistance as a function of inverse temperature with (green curves) and without (magenta curves) long-range interactions. Thick lines correspond to the case without site disorder, while thin lines to disordered samples.

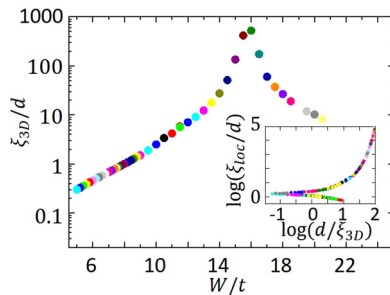

**Fig. 5 Evolution of the 3D localization length $\xi_{loc}$ with the disorder.** Each color corresponds to a given disorder $W$. $\xi_{3D}$ is the bulk localization length, $d$ is the sample thickness, and $t$ is the kinetic energy of the electrons. There is a localized−delocalized transition around a critical disorder $W_c \approx 16.5$. Inset: $\xi_{loc}/d$ as a function of the localization length that the corresponding bulk (3D) system would have, on a double logarithmic scale. The localized−delocalized transition occurs for $\log\left(\xi_{loc}/d\right) \simeq 0.3$.

of $Nb_xSi_{1−x}$ films, private communication). Most of this variation comes from the geometrical disorder, i.e., from the changes in the number of neighbors of each grain.

**Localization length**. For the calculation of the localization length, we consider the standard Anderson Hamiltonian for spinless particles

$$H = \sum_i \epsilon_i n_i + \sum_{\langle i,j \rangle} t(c_j^\dagger c_i + c_i^\dagger c_j), \qquad (16)$$

where $c_i^\dagger$ is the creation operator on site $i$ and $n_i = c_i^\dagger c_i$ is the number operator. We consider a transfer energy $t = −1$, which sets our unit of energy, and a disordered site energy $\epsilon_i \in [−W/2, W/2]$. The double sum runs over nearest neighbors. We have considered square samples of finite thickness with dimensions $L \times L \times d$. All calculations are done at an energy equal to 0.1, to avoid possible specific effects associated with the center of the band.

We have studied numerically the zero temperature conductance $g$ proportional to the transmission coefficient $T$ between two semi–infinite leads attached to opposite sides of the sample,

$$g = \frac{2e^2}{h} T, \qquad (17)$$

where the factor of 2 comes from spin. We measure the conductance in units of $2e^2/h$. We calculate the transmission coefficient from the Green function, which is obtained propagating layer by layer with the recursive Green function method[62]. We can solve samples with lateral section up to $L \times d = 400$. The number of different realizations employed is $10^4$ for most values of the parameters. We have considered wide leads with the same section as the samples, represented by the same Hamiltonian as the system, but without diagonal disorder. We use cyclic periodic boundary conditions in the long direction perpendicular to the leads, and hard wall conditions in the narrow traversal direction.

According to single-parameter scaling, the conductance is a function of the ratio of the two relevant lengths of our problem,

$$g = g_0 f(L/\xi_{loc}(W, d)),$$

where $L$ is the lateral system size and $\xi$ the correlation length, which carries the dependence on $W$ and $d$. Data for different $W$ and $d$ can be made to overlap by an adequate choice of $\xi_{loc}(W, d)$. We use this idea in both the diffusive and the localized regimes. In the former the conductance depends logarithmically on system size

$$g = g_0 − \frac{2}{\pi} \log\frac{L}{\xi_{loc}}, \qquad (18)$$

where the factor $2/\pi$ has been obtained by diagrammatic perturbation theory[63]. In the strongly localized regime, the conductance depends exponentially on the system size

$$g = c \exp\frac{−2L}{\xi_{loc}}. \qquad (19)$$

The results for $\xi_{loc}(W,d)$ are shown in Fig. 2d.

To analyze quantitatively the overall behavior of $\xi_{loc}$, we have performed a new one-parameter scaling analysis[62,64–66] of the data plotted in Fig. 2d. The idea is to plot $\log\left(\xi_{loc}/d\right)$ as a function of $\log\left(d\right)$ and shift the data horizontally by the disorder-dependent amount that best overlaps the data. This shifting quantity corresponds to the three-dimensional correlation length $\xi_{cor}^{3D}(W)$ which is either the

metallic correlation length if the bulk system with the same amount of disorder $W$ is delocalized or corresponds to the three-dimensional localization length $\xi_{loc}^{3D}(W)$ if it is localized (Fig. 5). The two branches therefore correspond to the well-known three-dimensional localization-delocalization transition at a critical disorder of about $W_c \approx 16.5$. The corresponding scaling is shown in the inset of Fig. 5. For small (resp. large) disorder levels, the system is in the upper (resp. lower) branch and the corresponding 3D system is delocalized (resp. localized): $\xi_{loc}$ increases (resp. decreases) faster than thickness. The transition between those two regimes, for our model, occurs when:

$$\log\left(\frac{\xi_{loc}}{d}\right) \approx 0.3. \qquad (20)$$

This implies that if for a given thickness and amount of disorder $\xi_{loc} > 2d$, the system will tend to metallic when its thickness increases and to an insulator when $\xi_{loc} < 2d$.

Usually in experimental situations, one does not know with enough precision the relation between $d$ and $\xi_{loc}$ to decide if a sample for a given disorder will become extended or localized when its thickness increases according to criteria (20). We can establish a new criteria to predict the 3D character of a system. If the 3D system corresponding to a given disorder value is extended, as its thickness decreases, the ratio $\xi_{loc}/d$ decreases, meaning that the localization length decreases faster than the thickness:

$$\frac{\xi_{loc,1}}{\xi_{loc,2}} > \frac{d_1}{d_2}, \qquad (21)$$

if $d_1 > d_2$. Conversely, if the 3D system corresponding to a given disorder value is localized:

$$\frac{\xi_{loc,1}}{\xi_{loc,2}} < \frac{d_1}{d_2}, \qquad (22)$$

for $d_1 > d_2$. Let us apply criterion (21) to our experimental samples.

For the most disordered $x = 13.5\%$ samples ($\theta_{ht} = 150\,°C$), one can estimate the ratio of the localization lengths through the relation $\xi_{loc} \propto T_0^{−1/2}$ (Eqs. (4) and (5)):

$$\frac{\xi_{loc,1}}{\xi_{loc,2}} \approx \sqrt{\frac{T_{02}}{T_{01}}} = 6 > \frac{d_1}{d_2} = \frac{4}{3} \qquad (23)$$

for $d_1 = 40\,Å$ and $d_2 = 30\,Å$ for instance. The corresponding 3D system is therefore on the metallic side of the metal−insulator transition. All considered $x = 13.5\%$ samples exhibit $\xi_{loc} \gg d$. The $x = 9\%$ samples are also extended, but closer to the transition: $\xi_{loc,1}/\xi_{loc,2} = 2.6$ for $d_1/d_2 = 2$.

## Data availability

All data in the main text or the Methods are available from the corresponding authors upon reasonable request. The analysis also includes data from refs. [15–17,22,23,50] (Shahar, D. Transport measurements of InOx films, private communication).

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

## Acknowledgements

The authors thank Zvi Ovadyahu for valuable discussions and guidance, and Dan Shahar's group for sharing their experimental data on InO$_x$. M.O. and A.M.S. acknowledge support by Fundación Séneca grant 19907/GERM/15 and AEI (Spain) grant PID2019-104272RB-C52. V.H., L.B., L.D. and C.A.M.-K. acknowledge the support of the ANR (grant ANR 2010 BLAN 0403 01).

## Author contributions

V.H., C.A.M.-K. designed and carried the experiments. V.H., C.A.M.-K., L.B and L.D. fabricated the devices. M.O. and A.M.S. performed the simulations. V.H., C.A.M.-K., M.O. and A.M.S. analyzed the data and wrote the manuscript.

## Competing interests

The authors declare no competing interests.
