## [Peer Review File · Nature Communications]

REVIEWER COMMENTS

Reviewer #1 (Remarks to the Author):

The authors present experimental results from Nb-Si films and results from Monte Carlo simulations investigating transport near superconductor-insulator transition (SIT). There is also a fair comparison with data from other material systems (Sn, Bi, InOx etc.).

The work is thorough and the details presented are clear. However, I have some comments about the conclusions:

(1) In homogeneous systems, the electronic granularity is a consequence of a diverging localization length and that the localization length depends exponentially on the film thickness. These are important findings and are useful to the SIT community.

(2) It is known that the region close to SIT that is governed by both superconducting gap and charging energy depends on grain size. It is not surprising.

(3) The most critical comment I have is that it is well-known now (due to many excellent works over last 10-15 years) that localized Cooper pairs exist on the insulating side of SIT. The nature of the novel insulating phase has been studied extensively by a variety of methods (transport, scanning techniques, high frequency methods etc.) and I do not see anything new in this conclusion.

In conclusion, it is a good work that includes data from a good set of samples and comparisons with other material systems. However, I do not see much novelty in the conclusions in this well-studied field.

Reviewer #2 (Remarks to the Author):

The problem of over-activated transport close to SIT is really very intriguing. It is also very exciting, as it may signal on the Many-Body localization which is a very hot subject at present. At the same time, it is true that this phenomenon has not been studied systematically, both experimentally and

theoretically. This manuscript is an attempt of such a study which should only be welcome. It is also important that the explanation of the vast experimental data presented in this manuscript is done using the usual, essentially classical, concepts such as charging energy or single-electron spectral gap which result in an energy cost for transport between superconducting islands.

The effect of the energy cost for a single electron to enter a superconducting island can in principle explain a super-activated behavior within a conventional model of islands connected by capacitors and at disorder induced by offset charges. The central verifiable result of the manuscript is Eq.(9) which proposes a relationship between the difference of characteristic energies T_0 and T'_0 in an activation law Eq.(1) for temperatures below and above the crossover one (which is associated with the superconducting transition temperature in an island).

As it was said above, the model is very simple and this is its definite advantage. It is clear that at temperatures lower than the superconductor transition temperature in a superconducting island there is an extra energy cost for transport and this energy cost manifests itself in an increased T'_0 . This indeed may be an explanation of "apparent super-activated behavior" in a number of cases.

However, probably, not in all the cases.

What is not discussed in this manuscript, however, is how to reconcile the behavior which is essentially a crossover between two activated behaviors with the well-known result of

Evidence for a Finite-Temperature Insulator.

Ovadia M., Kalok D. F., Tamir I., Mitra S., Sacepe B. & Shahar D. (2015) Scientific Reports. 5, 13503.

where the super-activated behavior was not a crossover between the two activated ones and the crossover temperature between the activated and super-activated behavior seems not to be related with the SIT:

Superconductor-Insulator Transition and the Crossover to Non Equilibrium in two-dimensional Indium - Indium-Oxide composite

Bar Hen, Xinyang Zhang, Victor Shelukhin, Aharon Kapitulnik, and Alexander Palevski,
arXiv:2003.05723

I think that those two works are directly related with the subject of the manuscript and they have to be cited and commented.

Another comment is that the superconducting gap is not homogeneous and vary a lot from island to island. Moreover, in more disordered samples the most probable gap may be even zero. This dispersion of the gap makes it difficult, within the model adopted in the manuscript, to attribute a definite crossover temperature. The distribution of the crossover temperatures significantly alters the form of the temperature dependence of the resistance of the whole sample. This effect should be considered or at least discussed.

In conclusion, the subject of the manuscript is important and timely. It contains a systematic experimental study of the phenomenon. The theoretical model is as simple as possible and the presentation is very well accessible to non-specialists. This material definitely adds up to the understanding of the super-activation phenomenon and its relation with the SIT. However, in my opinion, the most interesting experiments where the super-activation was observed are not explained by this simple model. If authors think differently, the discussion including the two above references is, in my opinion, necessary, as well as the discussion on the inhomogeneity of the energy gap.

Reviewer #3 (Remarks to the Author):

The present manuscript describes experiments and model calculations that reveal a stronger-than-Arrhenius behavior of the temperature-dependent resistivity of insulating NbSi-films. This behavior, dubbed "overactivation" is explained in terms of a cross-over from one activation temperature to another by the freezing of single-particle excitations. Numerical simulations can explain the observed exponential dependence of the localization length on both disorder and film thickness.

The work appears technically sound, both in experiment and theory. However, the wording is often very unprecise and there are a number of issues I request the authors to comment on and improve, respectively:

1) Preceding Eq. 4 an 'exponential gap' is mentioned - what does that mean? Exponential activation across the gap E_0 ? Please formulate clearly.

2) In Eq.4 the quantity κ_0 is only explained at the end of the next column much later.

3) I do not understand the statement: "Nevertheless, the introduction of disorder screens out the long-range contribution to the activation energy, so that r_{\max} corresponds to the inter-impurity distance.

'Screening out the long-range contribution' I would understand as a limitation of the range of Coulomb interactions by polarization charges - why should simple disorder do that? I would expect $r_{\max} = \kappa d$, as mentioned after Eq. 3. Which numbers of κd and r_{\max} are obtained? I would expect, that the average inter-impurity distance is not far from the typical interatomic distance a in such highly disordered films.

4) What is meant by " T_0 is smaller than the exponential behavior" - is it " T_0 deviates from the exponential behavior (solid line)"? But the data are on the solid line?? Can the large value $\kappa_0 = 250$ be reproduced by the calculation? Where does it physically come from?

How big can the second contribution $\kappa - \kappa_0$ become, according to this analysis?

5) In section 4 the observable T_{cross} seems the phenomenological crossover temperature between the two activated regimes. Afterwards, T_{cross} is identified with the superconducting transition temperature T_c of the isolated granules. That is plausible, but should then be systematically used, also in Eq. 9.

In Eq. 8 both T_c and T_{cross} appear simultaneously, as if they were different. That is very confusing!

6) It is unclear, how Eq. 10 is reflected in Fig.3d. Why are the two traces of (open and full) green squares rather than only one trace, depicting $T'_0 - T_0 = \gamma T_{\text{cross}} / \gamma T_c$? The sentence following Eq. 10 is explained only a paragraph later. It is incomprehensible after Eq. 10, and remains so, because the explanation in the following paragraph is incomprehensible too. Where is

Eq.10 coming from? Is there a theoretical reasoning, or is this purely empirical? What is new with respect to Refs. 47-52?

7) According to Eq. 8, Eq. 10 can be valid in the limit $T \rightarrow 0$ only. It seems that Eq. does not use the usual BCS-dependence of $\Delta(T)$, but only its limiting form valid near T_c (even incorrect - if the unspecified "BCS-ratio" means $\Delta/kT_c=2$, a prefactor 1.74 (Eq. 3.52 in Tinkham) seems to be missing before the root).

8) In Fig. 3d, one seems that T_{cross} can come very close to the bath temperature. How accurate is the procedure in this case?

9) Also the figures seem very inhomogeneous in terms of fonts and font sizes (sometimes hard to read). In addition, there are a number of unexplained comment (L,R). The arbitrary units in Fig. 3a are not good - if this is a scaled representation, say $R(T)/R(T_{\text{ref}})$ at some reference temperature T_{ref} , that should be made explicit and the figure with the unscaled data (Fig. 1e??) mentioned. Some error bars on the parameter γ in Figs. 3d,e should be included. The logarithmic scaling for this plot is not very justified, because the variations are rather small.

In conclusion, the paper is interesting, because it systematically studies the evolution of the activation temperature and the localization length separately as a function of disorder and thickness. It also offers a potential interpretation of "overactivation". However, many imprecisions are disturbing and the most important section 4 is very poorly written. Hence, I cannot recommend acceptance of the manuscript in its present form.

Answers to the Reviewers

“Overactivated transport in the localized phase of the superconductor-insulator transition”

Submitted to Nature Communications (NCOMMS-21-02570-T)

We thank all three referees for their careful reading and comments on our manuscript.

Reviewer #1:

The authors present experimental results from Nb-Si films and results from Monte Carlo simulations investigating transport near superconductor-insulator transition (SIT). There is also a fair comparison with data from other material systems (Sn, Bi, InOx etc.).

The work is thorough and the details presented are clear. However, I have some comments about the conclusions:

(1) In homogeneous systems, the electronic granularity is a consequence of a diverging localization length and that the localization length depends exponentially on the film thickness. These are important findings and are useful to the SIT community.

Answer:

We thank Reviewer #1 for his/her appreciation of our work. We indeed believe that an **investigation of the influence of the film thickness on the localization length** was needed, and that this provides an explanation for the electronic granularity that arises close to the superconductor-insulator transition (SIT). It is, indeed, one of our main results.

(2) It is known that the region close to SIT that is governed by both superconducting gap and charging energy depends on grain size. It is not surprising.

(3) The most critical comment I have is that it is well-known now (due to many excellent works over last 10-15 years) that localized Cooper pairs exist on the insulating side of SIT. The nature of the novel insulating phase has been studied extensively by a variety of methods (transport, scanning techniques, high frequency methods etc.) and I do not see anything new in this conclusion.

In conclusion, it is a good work that includes data from a good set of samples and comparisons with other material systems. However, I do not see much novelty in the conclusions in this well-studied field.

Answer:

Indeed, it is not surprising that both the superconducting gap and the charging energy depends on the grain size. However, in this work, we are being more specific and determine the exact role of both parameters (equations (9) and (10) of the manuscript).

Moreover, until now, there have been two main scenarii to explain the SIT. The bosonic picture has been put forward to explain the behavior in -at least electronically- inhomogeneous systems, such as InOx (Sacepe PRB 91, 174505 2015) or TiN (Baturina PRL 99, 257003 2007). For other thin films, such as NbN (Noat PRB 88, 014503 2013), MoC (Szabo PRB 93, 014505 2016), or MoGe (Gantmakher Physica C 404, 176 2004), the fermionic picture was thought to be the driving phenomenon at play. Our material, a-NbSi, has been a candidate for such a fermionic scenario (Feigel'man Ann. Phys. 325, 1390 2010, Gantmakher Physica C 404, 176 2004), since the typical signatures of the bosonic scenario (notably the giant magneto-resistance peak and the strong insulating features) have not been observed to date in this material. One would therefore not expect the superconducting gap to survive in the insulating regime for this system.

What we show here is that electronic granularity and persistence of Cooper pairs are also relevant for this type of systems. In other words, the presence of incoherent Cooper pairs in the insulating regime does not always result in giant magneto-resistance peaks for instance.

Moreover, the referee refers to the large corpus of experimental work existing on the SIT, measuring either inhomogeneities of the order parameter close to the SIT (but on the superconducting side) by STM measurements (Sacepe Nat. Phys. 7, 1892 2011 for instance) or by tunnel barrier measurements (Sherman PRB 89, 035149 2014) as well as high frequency measurements close to the SIT (Liu PRL 111, 067003 2013, Mondal Sc. Rep. 3, 01357 2013). While these techniques measure the nature of the system ground state, they do not allow to conclude on the nature of the excitations that are dominant for low frequency transport, and do not deal with the activated regime close to the SIT. These issues we address in the manuscript.

In addition to this, until now, there has not been any systematic experimental investigation of the over-activated regime. This kind of behavior had only been seen (to our knowledge) in TiN (Baturina JETP Lett 85 752 2008 and Baturina PRL 99 257003 2007) and InOx (Shahar's group, unpublished data), i.e. in materials thought to exhibit a bosonic scenario, and **under magnetic field**. We provide a new system in which this over-activated behavior is observed in zero-field, and analyze its appearance as a function of disorder. We believe it is important that the over-activated regime is observed, to our knowledge for the first time, in zero-field. This clearly signifies that magnetic field is not needed to explain this behavior. Moreover, no theoretical explanation has been proposed for this behavior until now, and we submit one possible scenario. **We have modified the text to clarify this last point.**

Reviewer #2:

The problem of over-activated transport close to SIT is really very intriguing. It is also very exciting, as it may signal on the Many-Body localization which is a very hot subject at present. At the same time, it is true that this phenomenon has not been studied systematically, both experimentally and theoretically. This manuscript is an attempt of such a study which should only be welcome. It is also important that the explanation of the vast experimental data presented in this manuscript is done using the usual, essentially classical, concepts such as charging energy or single-electron spectral gap which result in an energy cost for transport between superconducting islands.

The effect of the energy cost for a single electron to enter a superconducting island can in principle explain a super-activated behavior within a conventional model of islands connected by capacitors and at disorder induced by offset charges. The central verifiable result of the manuscript is Eq.(9) which proposes a relationship between the difference of characteristic energies T_0 and T'_0 in an activation law Eq.(1) for temperatures below and above the crossover one (which is associated with the superconducting transition temperature in an island).

As it was said above, the model is very simple and this is its definite advantage. It is clear that at temperatures lower than the superconductor transition temperature in a superconducting island there is an extra energy cost for transport and this energy cost manifests itself in an increased T'_0 . This indeed may be an explanation of "apparent super-activated behavior" in a number of cases.

However, probably, not in all the cases.

What is not discussed in this manuscript, however, is how to reconcile the behavior which is essentially a crossover between two activated behaviors with the well-known result of

- "Evidence for a Finite-Temperature Insulator" by Ovadia M., Kalok D. F., Tamir I., Mitra S., Sacepe B. & Shahar D. (2015) Scientific Reports. 5, 13503. where the super-activated behavior was not a crossover between the two activated ones
- and the crossover temperature between the activated and super-activated behavior seems not to be related with the SIT: "Superconductor-Insulator Transition and the Crossover to Non Equilibrium in two-dimensional Indium - Indium-Oxide composite" by Bar Hen, Xinyang Zhang, Victor Shelukhin, Aharon Kapitulnik, and Alexander Palevski, arXiv:2003.05723

I think that those two works are directly related with the subject of the manuscript and they have to be cited and commented.

Answer:

We thank Reviewer #2 for his/her appreciation of our work. We do indeed think that over-activation has often been talked about, but that there are very few peer-reviewed works, apart from those mentioned in the manuscript. We also think the path of digging into usual concepts to explain this phenomenon is interesting and should not be overlooked.

First, let us note that both works mentioned by the referee, which we have now included in the manuscript, involve magnetic fields and that the situation may then be more complicated. Moreover, our results may not apply to all systems and particularly those where the overactivated behavior is observed only in the presence of magnetic fields. **We have clarified this in the manuscript.**

In our manuscript, the InOx data in figure 3.e were already data from Dan Shahar's group (30 nm-thick InOx sample "RAM005b_034_R", ref 47 in the manuscript). These are similar to the reference the referee suggested (Ovadia, Sc. Rep. 5 13503 2015). The "RAM005b_034_R" sample exhibits a stronger than activation behavior at low temperature under magnetic field. Below is the corresponding $R(1/T)$ plot. As can be seen, this film can very well be explained by two activated regimes.

Figure 1 Data from the 30nm-thick InOx sample "RAM005b_034_R" (ref 47 in the manuscript) used in the analysis of figure 3.d.

As for the samples investigated in Ovadia, Sc. Rep. 5 13503 2015, it is difficult to conclude from the data that have been published: in the low B regime where the stronger than activated behavior is observed, there are only 5-6 data points. However, the plot below is a re-analysis of the data presented in Fig.2 of the supplementary information of Ovadia (data taken from the figure in the paper). It shows that the data is compatible with two distinct activation regimes with: $T_0 = 0.1$ K, $T'_0 = 0.4$ K and $T_{cross} = 0.06$ K, which yields $\gamma \approx 5$, with the notation of figure 3 and eq. 9, in fair agreement with our prediction. **We included this analysis in Fig 3.e.**

Figure 2: Re-analysis of the $R(T)$ data presented in Fig.2 of the supplementary information of *Ovadia Sc. Rep.*, 5 13503 2015. The sample is 30 nm-thick InOx under a 0.75 T magnetic field.

We thank Reviewer #2 for drawing our attention to the second reference (Hen PNAS, 118 2015970118 2021). **We have included it as a reference in the manuscript.** If we consider figure 5 (reproduced below), the red points signal the crossover between the activated and the “super-activated” behaviors. This temperature corresponds to our T_{cross} .

Fig. 5. A rough determination of the crossover temperature from equilibrium VRH behavior to the nonequilibrium state of sample 1-second anneal. Black circles denote the first deviation from VRH with $\delta = 1/2$, whereas red circles denote the crossover to “superactivation” with $\delta > 1$.

Figure 3: Reproduction of Figure 5 from Hen PNAS, 118 e2015970118 2021.

We can reanalyze the data of this figure for the most insulating samples (the 110 mT and 112.5 mT data are available on the Github link given at the end of the article). This is shown in the figure below. First, we can see that these films are not very disordered, so that the “super-activated” regime only appears at the lowest temperature. As a consequence, the data points we can base our analysis are scarce.

Figure 4: Re-analysis of the $R(T)$ data presented in Fig.5 of Hen PNAS 118 2015970118 2021. The sample is 30 nm-thick InOx coated with In islands under magnetic fields of 110 and 112.5 mT.

From low temperature fits, we can infer *rough* estimates of the various quantities of our eq. (9):

B (mT)	T_0 (mK)	T'_0 (mK)	T_{cross} (mK)
110	31 (green fit)	37 (dark grey fit)	28
112.5	30 (blue fit)	39 (light grey fit)	23

These values give $\gamma \sim 0.25 - 0.4$, similar to what we obtain with our eq. (9).

Let us note that, in this system, the superconducting critical temperature T_c of the In grains is of about 3.4 K, whereas the T_c of the InOx layer is of about 1.5 K at zero field. However, beyond 90 mT (see figure 1 of Hen's paper), all signs of macroscopic superconductivity disappears, both at 1.5 K and at 3.4 K. There, T_{cross} cannot be linked to these T_c in a simple manner. T_{cross} reflects the temperature at which superconducting grains appear in the system. In the case of a system under magnetic field, this could be significantly different from the In or InOx T_c at zero field.

If the "super-activated" regime was linked to superconductivity in this system – which, again, is not obvious since it is a finite magnetic field experiment – one would expect a stronger "super-activated" regime at low field. From the data (a low temperature zoom of Hen's figure 5 is given below), this is unclear:

- The "super-activated" regime does not have enough data for us to conclude
- The low field data are a bit noisy
- Below 40 mK, the data does not evolve systematically with magnetic field, rendering any definite conclusion on the low temperature regime dubious.

Figure 5: Zoom of the low temperature part of figure 5 of Hen PNAS 118 2015970118 2021. The sample is 30 nm-thick InOx coated with In islands under magnetic fields of 110 and 112.5 mT.

Finally, let us stress that we are not saying that our scenario of the two activated behaviors accounts for what happens in all other systems, especially under magnetic field, but we would like to point out that this interpretation is certainly compatible with published data. **We have clarified this in the manuscript.**

Another comment is that the superconducting gap is not homogeneous and vary a lot from island to island. Moreover, in more disordered samples the most probable gap may be even zero. This dispersion of the gap makes it difficult, within the model adopted in the manuscript, to attribute a definite crossover temperature. The distribution of the crossover temperatures significantly alters the form of the temperature dependence of the resistance of the whole sample. This effect should be considered or at least discussed.

Answer:

We agree with the reviewer that there must be some dispersion in the values of the gap. Although we did not specify it in the first version of the manuscript, our simulation was as realistic as possible regarding this point. We considered a disordered array of grains and random capacitances between them (as explained in the Materials and Methods section) in order to get a distribution of charging energies, and so of gaps, similar to the distribution measured experimentally. Indeed, Claude Chapelier (INAC, Grenoble, France) had performed STM measurements on superconducting a-NbSi films ($\text{Nb}_{15}\text{Si}_{85}$, 50 nm, as deposited) and measured a dispersion of about 36%, as shown in the figure below (unpublished).

Figure 6: Gap distribution in a 50 nm-thick $\alpha\text{-Nb}_{15}\text{Si}_{85}$ sample, as measured by STM by C. Chapelier [unpublished].

In the figure below, we show our distribution of charging energies.

Figure 7: Charging energy distribution for our numerical simulations.

We have clarified this in the main text and in the Materials and Method section.

In conclusion, the subject of the manuscript is important and timely. It contains a systematic experimental study of the phenomenon. The theoretical model is as simple as possible and the presentation is very well accessible to non-specialists. This material definitely adds up to the understanding of the super-activation phenomenon and its relation with the SIT.

However, in my opinion, the most interesting experiments where the super-activation was observed are not explained by this simple model. If authors think differently, the discussion including the two above references is, in my opinion, necessary, as well as the discussion on the inhomogeneity of the energy gap.

Answer:

We thank Reviewer #2 for his/her appreciation of our work. We have tried to clarify the points he/she mentioned.

Reviewer #3:

The work appears technically sound, both in experiment and theory. However, the wording is often very unprecise and there are a number of issues I request the authors to comment on and improve, respectively:

1) Preceding Eq. 4 an 'exponential gap' is mentioned - what does that mean? Exponential activation across the gap E_0 ? Please formulate clearly.

Answer:

An exponential gap is a hard gap with a density of states increasing exponentially with energy. It is analogous to Efros and Shklovskii Coulomb gap for a logarithmic interaction. **We have briefly explained its meaning in the text and we have introduced a new reference (Somoza et al, 2015) where the exponential gap is properly explained.**

2) In Eq.4 the quantity κ_0 is only explained at the end of the next column much later.

Answer:

The definition of κ_0 has been inserted right after Eq. 5.

3) I do not understand the statement: "Nevertheless, the introduction of disorder screens out the long-range contribution to the activation energy, so that r_{\max} corresponds to the inter-impurity distance.

'Screening out the long-range contribution' I would understand as a limitation of the range of Coulomb interactions by polarization charges - why should simple disorder do that? I would expect $r_{\max} = \kappa d$, as mentioned after Eq. 3. Which numbers of κd and r_{\max} are obtained? I would expect, that the average inter-impurity distance is not far from the typical interatomic distance a in such highly disordered films.

Answer:

We agree with the referee that the wording we used could be misleading, and **we have changed it to try to avoid any misunderstanding**. Polarization charges will be present and their effect is taken into account through the value of the dielectric constant κ . For a given dielectric constant, if there are charges in the system, the effective range of the logarithmic interaction will be roughly the minimum between ($\kappa \times$ thickness) and the distance between charges (as there will be screening between charges). Disorder introduces charges into the system which decrease this maximum range of the logarithmic interaction. We referred to inter-impurity distance because our way to introduce disorder corresponded to inserting $\pm e$ charges at some fixed random positions. In general, we expect that disorder will help to introduce charges into the system and we argue that these charges will cut out the logarithmic dependence of the potential at a distance r_{\max} of the order of the distance between charges.

In the manuscript, we have explained the corresponding paragraph with some more detail.

The films are quite disordered relative to the clean systems, but their localizations lengths are still huge.

The minimum T_0 is of the order of 0.1 K, which corresponds to values of κd approx 10^7 \AA . r_{max} should be an effective impurity distance, which is difficult to calculate, but only enters in a logarithm and its influence is small. All we are saying, in practice, is that the activation temperature is equal (in units of Boltzmann constant) to the charging energy given by Eq. 5 with the logarithm playing a very minor role.

4) What is meant by "T_0 is smaller than the exponential behavior" - is it "T_0 deviates from the exponential behavior (solid line)"? But the data are on the solid line?? Can the large value $\kappa_0=250$ be reproduced by the calculation? Where does it physically come from? How big can the second contribution $\kappa - \kappa_0$ become, according to this analysis?

Answer:

We have rephrased the manuscript, which was misleading, to clarify this point.

We did not include the expression of the solid line due to length limitations: T_0 deviates from the exponential behavior, which corresponds to a straight line in Fig. 2c. The solid line is a fit of the data to $T_0 = \frac{1}{A+B\exp(C*d)}$ which comes from Eqs. 4 and 5. The constant A comes from the host dielectric constant κ_0 . From this fit we obtain $\kappa_0 \approx 250$. κ_0 is not calculated, it is obtained from the fit of the activation temperatures.

Note (figure below) that κ_0 does not have a large influence on the fit, except at low thicknesses. Its precise determination is therefore difficult, but we can estimate it to range between about 150 and 350. Such value is not surprising. Indeed, low temperature dielectric constant of bulk amorphous silicon can be of the order of 20 (Aspnes PRB 29 768 1984) and the proximity to the SIT should increase this value (Wu ArXiv/0511121).

The second contribution $\kappa - \kappa_0$ can be up to four orders of magnitude larger than κ_0 , so we get a maximum value of κ of the order of 10^6 .

Figure 8: Influence of the fitting parameter κ_0 on the fit of figure 2c of the paper.

5) In section 4 the observable T_{cross} seems the phenomenological crossover temperature between the two activated regimes. Afterwards, T_{cross} is identified with the superconducting transition temperature T_c of the isolated granules. That is plausible, but should then be systematically used, also in Eq. 9. In Eq. 8 both T_c and T_{cross} appear simultaneously, as if they were different. That is very confusing!

Answer:

Thank you for noticing this. **We have unified our notation and replaced T_c by T_{cross} it in two places.**

6) It is unclear, how Eq. 10 is reflected in Fig.3d. Why are the two traces of (open and full) green squares rather than only one trace, depicting $T'_0 - T_0 = \gamma T_{cross} / \gamma T_c$? The sentence following Eq. 10 is explained only a paragraph later. It is incomprehensible after Eq. 10, and remains so, because the explanation in the following paragraph is incomprehensible too. Where is Eq.10 coming from? Is there a theoretical reasoning, or is this purely empirical? What is new with respect to Refs. 47-52?

Answer:

The predictions of Eq. (9), $T'_0 - T_0 = \gamma T_{cross}$, are the orange triangles. Green squares refer to Eq. (10).

We have rewritten the two paragraphs relative to Eq. (10) to explain it better. This equation is an empirical interpolation between two well-known limits. Ref. 47-53 only deal with the first term in the RHS of Eq. (10), the $1/d$ dependence, while Ref. 54 only refers to the second term. What is new here is the interpolation between the two limits.

In the previous version, the solid squares represented the T_{cross} when only the over-activated behavior was observed while the empty squares referred to the situation where two activated behaviors were observed. The difference between the two was that one could be compared with the

experimental T_{cross} while the other one could not since only one regime was observed and there was no crossing temperature. In our new iteration of the manuscript, we chose not to distinguish between these two regimes for clarity.

7) According to Eq. 8, Eq. 10 can be valid in the limit $T \rightarrow 0$ only. It seems that Eq. does not use the usual BCS-dependence of $\Delta(T)$, but only its limiting form valid near T_c (even incorrect - if the unspecified "BCS-ratio" means $\Delta/kT_c=2$, a prefactor 1.74 (Eq. 3.52 in Tinkham) seems to be missing before the root).

Answer:

One of the aims of the simulations was to show that the spread in charging energies smooths the singularity in the superconducting gap given by Eq. (10) so that a fairly well-defined activated regime is obtained. The calculations showed that this was indeed the case, as can be better appreciated in Fig. 3c. Near T_c should be a drastic change, but it is compensated by the dispersion in grain sizes and so in transition temperatures.

The BCS ratio we have taken corresponds to the one that has been measured by STM by Claude Chapelier on superconducting a-NbSi films (our reference 49). Indeed, in thin disordered films, the BCS ratio of 1.76 is not always obeyed (see for instance Sacepe Nature Physics 7 1892 2011, our reference 8). **What was probably misleading in the text was the reference to BCS theory, which we modified.**

8) In Fig. 3d, one seems that T_{cross} can come very close to the bath temperature. How accurate is the procedure in this case?

Answer:

Let's take the example of a film close to the SIT (13.5%, 40A, 90C). As can be seen in this figure, two simply activated regimes are well defined. T_0 is between 0.18K and 0.2K and T_0' between 0.25K and 0.26K. The crossing temperature is between 0.065 and 0.08, close to the bath temperature. This results in γ between 0.625 and 1.23. This strong uncertainty is mostly due to the limited range of the fit, which is especially large close to the SIT due to restricted temperature ranges. This is probably the reason why γ deviates from its median value at low disorder. We would like to note however that the systematics of the data analysis for all our films allowed us to deduce a median γ parameter taking into account samples for which the over-activated regime is sufficiently large to reduce the uncertainties.

Figure 9 Re-analysis of a film of α -NbSi (13.5%, 40Å, 90C) used for extracting data figure 3.d. The film is close to the SIT ($R_{200K}=3940$ Ohms) and shows two well defined simply activated behaviors.

9) Also the figures seem very inhomogeneous in terms of fonts and font sizes (sometimes hard to read). In addition, there are a number of unexplained comment (L,R). The arbitrary units in Fig. 3a are not good - if this is a scaled representation, say $R(T)/R(T_{ref})$ at some reference temperature T_{ref} , that should be made explicit and the figure with the unscaled data (Fig. 1e??) mentioned. Some error bars on the parameter γ in Figs. 3d,e should be included. The logarithmic scaling for this plot is not very justified, because the variations are rather small.

Answer:

We did modify the figures to get more homogeneous fonts and font sizes and explained missing notations (namely L and R, which were explained in the methods but not in the figure captions).

Fig. 3a highlights the discrepancy between the low temperature behavior and the simply activated Arrhenius law usually describing the low temperature regime. Renormalization is performed by renormalizing the temperature axis by $T^* = T/T_0$ and collapsing all the resistances at a temperature large enough (around $T = 1.65 T_0$) so that the highest temperature simply activated regime is well defined. **We removed the logarithmic scaling for γ in Figs 3 d,e and added error bas.**

In conclusion, the paper is interesting, because it systematically studies the evolution of the activation temperature and the localization length separately as a function of disorder and thickness. It also offers a potential interpretation of "overactivation". However, many imprecisions are disturbing and the most important section 4 is very poorly written. Hence, I cannot recommend acceptance of the manuscript in its present form.

Answer:

We have modified the manuscript substantially and hope it is now clearer.

REVIEWERS' COMMENTS

Reviewer #2 (Remarks to the Author):

I carefully read the revised manuscript and I am fully satisfied by the explanations of authors and there amendments to the text.

Reviewer #3 (Remarks to the Author):

The authors have responded adequately to the points raised in the previous reports and the manuscript is written much clearer now. I think that this comprehensive study elucidating the role of disorder in the explanation of overactivated behavior is of wide interest. I recommend publication of the manuscript.